# Evidence of Disaster Planning by Home Care Providers: An Integrative Literature Review

**DOI:** 10.3390/ijerph20095658

**Published:** 2023-04-27

**Authors:** Sanne Lessinnes, Michael Köhler, Michael Ewers

**Affiliations:** Charité–Universitätsmedizin Berlin, Corporate member of Freie Universität Berlin and Humboldt Universität zu Berlin, Institute of Health and Nursing Science, Augustenburger Platz 1, 13353 Berlin, Germany

**Keywords:** home care providers, disaster preparedness, disaster planning, evidence

## Abstract

The increasing risk of disasters worldwide poses challenges both to health care infrastructures and to home care providers, who must maintain decentralised services for those in need of long-term care for as long as possible, even under adverse circumstances. However, both the kind of organisational precautions that home care providers consider in preparation for disasters and the available evidence on the effectiveness of these precautions remain largely unclear. An integrative literature review was thus performed via a systematic search of several international databases in order to identify original research on organisational disaster planning by home care providers and to determine the evidence base of this research. The quality of the included studies was assessed using the Mixed Methods Appraisal Tool. Of the 286 results, 12 articles met the inclusion criteria and presented results from nine studies on disaster planning. Three overarching types of activities carried out by home care providers were identified inductively. The overall scientific quality of the studies was moderate, and none investigated the effectiveness of disaster planning by home care providers. Despite the variety of possible activities that home care providers already consider, evidence on how to make organisational disaster planning effective and sustainable remains lacking.

## 1. Introduction

Both the risk of various kinds of disasters and the multifaceted consequences of these events are expected to increase around the world in the coming years [1]. Despite having long been spared from such developments, Europe—and especially Germany—is now also increasingly affected by these risks due, for example, to progressing climate change, the unsustainable use of natural resources, degrading ecosystems, pandemics, and epidemics [2]. In 2015, the United Nations addressed this global challenge by adopting the Sendai Framework for Disaster Risk Reduction 2015–2030, which is an international agreement aimed at significantly reducing both damage to critical infrastructure and the disruption of essential services [3]. The framework aims to reach this goal via improved disaster management both in and between all sectors as well as via activities that are targeted at disaster preparedness at the national, local, and organisational levels. In order to implement the Sendai Framework, a German strategy for strengthening resilience to disasters was developed and published in 2022 [4].

One essential component of this German resilience strategy is to better prepare critical health care infrastructures. Although attention in this context is often directed towards acute and emergency care and hospitals, home care should not be forgotten. In 2016, 20% of households in the European Union with individuals who required assistance used some sort of professional home care service [5]. Demand for home care can be assumed to further increase in the future in connection with demographic, epidemiological, and societal developments, as well as via the encouragement of health and social policy initiatives [6]. The ongoing shift of patient groups and forms of treatment (e.g., mechanical ventilation, haemodialysis, palliative care) from the inpatient sector to patients’ homes is also contributing to the growing importance of home care [7,8,9]. Independent of the vast differences in provision structures as well as the role of skilled professionals and regulatory mechanisms [10], home care is an indispensable part of the health and social care systems in many countries. It is therefore critical to better prepare this vital infrastructure for future disasters.

As members of community-based organisations, home care providers play a central role in identifying hazards, developing preparedness plans, responding to disasters, disseminating information, and identifying vulnerable individuals [11]. People in need of home care—and especially those who are immobile, disoriented, severely ill, or technology-dependent—are exposed to special risks during disasters that include large-scale power outages, floods, storms, and epidemics. The vulnerability of this population has been demonstrated in previous disasters and studies. For example, deficits in preparedness, disruption of services in the event of a disaster, or reduced ability and preparedness to evacuate from their homes can lead to serious negative health impacts for people who rely on home care [7,8,12]. In order to ensure the safety of these vulnerable individuals and to maintain their decentralised care for as long as possible, even under adverse circumstances, home care providers and their staff need to be organisationally prepared and ready for action at all times [9,13]. Indeed, they must be able to work closely with other health care providers as well as with emergency services and disaster response teams in the event of an incident, and they must be prepared to initiate any necessary evacuation. In the United States (henceforth, USA), a regulation was established in 2016 that includes national disaster requirements for Medicare- and Medicaid-participating providers in order to ensure adequate planning for disasters and to coordinate with emergency preparedness agencies [14]. However, it remains largely unclear what organisational arrangements home care providers have made or will make to prepare for disasters.

Aside from a few anecdotal reports in the literature (primarily from the USA) suggesting that home care providers are currently insufficiently prepared for disasters [13,15,16,17,18], as well as the availability of a literature review of best practises for disaster planning by home care providers (also from the USA) [19], it is unknown whether preparedness requirements are systematically implemented by home care providers or how these requirements impact the providers’ disaster preparedness. Thus, there is a lack of knowledge not only about the organisational precautions that home care providers consider or implement in preparation for disasters but also about how effective the activities that have already been initiated are as well as what evidence they are based on. The present research was designed to address this knowledge gap. Therefore, the aim of the present study was to provide information from the international literature about (i) organisational precautions that home care providers consider in preparation for disasters, (ii) whether these measures are systematically implemented, and (iii) what is known about the effectiveness of the measures. Based on this information, the goal was to form recommendations for disaster preparedness in this field, if possible. 

The work is part of a multiphase empirical research project on the “Maintenance of home care infrastructures in crisis and disasters” (AUPIK), which runs from February 2020 to June 2023 and is funded by the German Federal Ministry of Research (BMBF) in the context of the civil security research programme (SOFI). AUPIK aims to gain knowledge about how to increase the disaster resilience of home care infrastructure in Germany. In addition to conducting empirical surveys and developing specific tools in support of disaster preparation among home care providers, the objectives of the project also include examining the international state of research and the evidence base on this topic.

## 2. Methods

An integrative literature review was carried out in order to identify (i) original research on organisational disaster planning activities that are carried out by home care providers and (ii) the documented effects of these activities. This approach enabled different designs to be combined in order to uncover international evidence of innovative activities [20,21].

A systematic literature search was conducted in PubMed (including Medline), CINAHL, and Cochrane. Reference lists of eligible articles were also screened for suitable articles. The SPICE model [22] served as the starting point for defining search components and identifying search terms. The model was used in a simplified form with two components due to the research question: The component of “setting” comprised terms regarding home care providers, while the combined component of “intervention and evaluation” was related to measures for disaster preparedness and (if applicable) the evaluation of these measures. Table 1 lists the search commands for identifying relevant studies, including medical subject heading terms (MeSH) and title and/or abstract screening terms (TIAB).

Articles were included in the review if they represented original primary research studies. Measures of disaster preparedness had to be taken at the organisational level. There were no restrictions placed on the type of disaster. A data limitation was set for up to the year 2001 because the attacks on the World Trade Centre as well as the anthrax incidents in the USA—both of which occurred in 2001—increased the awareness of and efforts to improve disaster preparedness in many countries [23,24]. Studies that focused on other health care providers or settings, on educational aspects at the individual level only, or exclusively on the COVID-19 pandemic were excluded, as were articles without research value.

Thematic analysis of the included articles was carried out as a systematic procedure—following Whittemore & Knafl [21] —along the five steps of data reduction, data display, data comparison, conclusion, and verification. During data reduction, two researchers (SL, SG) extracted data independently, and discrepancies were discussed with a third member (MK) of the research team. Subsequently, the data from the studies were clustered and presented as tables using an inductive categorisation system.

The Mixed Methods Appraisal Tool (MMAT) [25] was used to critically appraise the included papers. The MMAT is a well-evaluated checklist [26] and is suitable for appraising qualitative, quantitative, and mixed-methods studies alike.

## 3. Results

### 3.1. Results of the Literature Search

The search initially yielded 286 articles. After screening, 25 articles were identified for closer inspection. These articles were independently reviewed by two researchers (SL, SG) via a full reading. A third member of the research team (MK) was consulted for making the decision as to whether to include an article in cases of doubt. Finally, 12 articles remained that met the inclusion criteria. The results of the electronic search were entered into the literature management programme EndNote (Version X9.2 for Windows). Figure 1 presents the search results as a flow chart based on the PRISMA statement [27].

### 3.2. Characteristics of the Included Studies

All 12 considered articles originated from the USA and were published between 2003 and 2020 (Table 2). One follow-up study was documented in three different articles [28,29,30]. Another empirical study was published in two articles [31,32]. As a result, nine different original research studies were included in this analysis.

### 3.3. Thematic Analysis

The thematic analysis inductively identified three overarching areas of organisational disaster planning by home care providers: (1) internal operational arrangements, (2) information and networking activities, and (3) measures for safeguarding patient care. These three areas are presented here in condensed form.

#### 3.3.1. Internal Operational Arrangements

Based on the studies considered here, home care providers’ internal operational arrangements for disaster planning were assigned to three subcategories, as shown in Table 3.


*Organisational disaster plan*


Disaster plans were mentioned relatively often as an organisational measure that home care providers use to prepare for disasters [28,29,30,31,32,33,34,35,36,37,39]. These plans covered, for example, an incident command system as well as other communication plans, a hazard vulnerability analysis, employee protection, an integrated business continuity plan, processes for safeguarding client records, material resources, agreements with partners, and other procedures for patient tracking. Information on whether these plans were regularly reviewed, tested in exercises, or even adapted after specific disasters was rare. The plans presented differed between the included articles and the agencies interviewed in them. For example, some of the agencies reported that their plans essentially consisted of maintaining a list of all patients and their caregivers with emergency contact information [35]. Another article presented a toolkit for creating a plan that takes an all-hazards approach and addresses the factors that home care providers need to be prepared for [37].


*Staff disaster arrangements*


Less frequently, home care providers mentioned measures for supporting the willingness and readiness of employees to work even under adverse circumstances and in times of disaster [28,29,30,36]. Such measures included, for example, the provision of individual disaster plans for staff members, securing child or adult care arrangements, keeping emergency supplies or emergency “go-kits” in employees’ homes, and having a pre-arranged meeting point for employees in the event of an emergency (e.g., a blackout or communication failure). In one study, a respondent reported that personal back up emergency plans are in place for staff relatives to ensure continuity of services for all patients [28,29,30].


*In-house disaster training*


Internal disaster training for the staff of home care providers ranged from no or minimal training to routine and comprehensive training that was offered either as initial training upon hire, as annual refresher courses, or as targeted training for different staff depending on their role in the agency or in the disaster plan [28,29,30,33,34,35,37]. Themes and didactic approaches were diverse and comprised, for example, online systems or platforms for delivering learning modules, staff meetings, “lunch and learns”, local disaster planning meetings for discussing certain topics, simulations, tabletop drills, and annual mock drills with home care staff for testing policy protocols. One agency surveyed found that conducting mock drills regularly each year resulted in a more orderly response (e.g., staff were more prepared, patients were better able to adapt) [35].

#### 3.3.2. Information and Networking Activities

This category included any activities that the home care providers in the included studies mentioned for communicating disaster-related information in preparation for—as well as for communication during and after—a disaster affecting their staff (internal), the users of their home care services, and other providers (external), as well as for networking with other organisations or providers in their region (see Table 4).


*Disseminating information to staff (internally)*


Electronic communication tools such as the telephone, text messaging, and email were often mentioned as a means of spreading disaster-related information. Some providers used secure text messaging systems, software, or mobile apps, while others relied on manual telephone chains or call trees. Sometimes, backup strategies for employee accessibility during communication outages were mentioned, such as an 800-Hz-radio battery-operated walkie-talkie system, web-based scheduling, two-way digital radio communication devices, or a contract with local ham radio operators [28,29,30,31,32,33,35,36].


*Disseminating information to home care users*


Identifying emergency contact information and informing patients and caregivers about disaster preparedness can be part of the patient admission process and may be reviewed regularly. Some providers communicated with patients via telephone after an incident in order to ensure the patients’ safety, to see if the patients needed anything, or to check whether a family member was available to help them [28,29,30,31,32,33,34,35,36,38,39]. An alternative mode of communication was “knocking on doors” to assess patients’ status in case of a communication disruption [28,29,30]. 


*Disseminating information to others (externally)*


Emergency notifications from a local or state emergency management or public safety agency, as well as health alerts from a local or state health department, were mentioned by some home care providers as important sources of external information. Other providers faxed or forwarded patient lists containing information such as medical risk, supply reserves, transportation needs, contact information, and special needs to local emergency responder agencies [28,29,30,33,35,38,39]. A study reports on the New York State Health Provider Network (HPN), a web-based system for exchanging information quickly and efficiently. It includes a communications directory with contact information and a health alert network. The HPN coordinator is an individual who has a designated role in each agency and serves as the primary point of contact [28,29,30].


*Networking*


Networking initiatives by home care providers can be directed to the local government, law enforcement, public health departments, emergency management agencies, the Red Cross, the EMS, the local fire department, the transportation industry, nursing homes, or hospitals. Informal agreements that specify roles or provide support with emergency responders, supply companies, or other home care providers were also mentioned, as was informal advice from local authorities, for example, on disseminating literature on preparedness, setting up a website, or organising awareness-raising events [28,29,30,31,32,33,34,35,36,37]. One article noted that agency cooperation and collaboration with local, regional, or state preparedness partners varied (from active participation to the inability to find partners). However, many agencies interviewed indicated that this collaboration is challenging [33].

#### 3.3.3. Measures for Safeguarding Patient Care

This category summarises the organisational disaster planning measures of home care providers that were designed to ensure patient care in the event of a disaster (see Table 5).


*Personal patient disaster plan*


In order to develop individual disaster plans [31,32,33,36,38] with which patients could ensure their own care, one measure that was mentioned in some of the studies considered here involved reviewing these plans regularly and adjusting them after a disaster [31,32,33]. 


*Disaster education*


Disaster education for patients and relatives was also mentioned as a preparatory measure that was mostly conducted during patient admission. However, wide variation was found in terms of the scope of the training and the covered topics (e.g., discussions of the emergency plan, emergency contact information, emergency shelter information, evacuation procedures, patient responsibilities, emergency kits, medical equipment, and patient safety) [31,32,33,34,35,36,38]. Occasionally, leaflets and written instructions were simply handed out [35,36]. A study presented and evaluated a checklist-like assessment tool to help providers assess and train their patients’ disaster preparedness [37].


*Triage/patient classification*


Classifying patients into risk-category groups in order to prioritise care during disasters was another way that home care providers prepared for an incident. Two-, three-, four-, or five-level categorisation systems were applied based on selected characteristics, which included the time frame in which each patient needed to be seen, the patient’s medical needs, and the support available to the patient [28,29,30,31,32,33,34,36,38,39]. In some cases, providers reported continually updating their rankings, especially in preparation for predicted events [33,39]. Based on the experiences of the interviewed agencies with their systems, the authors of one selected article developed a standardised classification system [39].


*Evacuation preparedness*


Some home care providers prepared an evacuation plan upon patient admission, assisted their patients in registering for emergency transport and shelter, and educated them about the evacuation procedure [31,32,33,34,35,36,38,39]. Other providers relied on Emergency Medical Service (EMS) for patient evacuation but were willing to provide medical and nursing care at the evacuation site or in special-needs shelters [34]. In some cases, home care providers tracked their patients after evacuation [31,32,33]. In one study, agencies had different degrees of involvement with special-need shelters (e.g., some were actively involved, others hardly at all) [35].

### 3.4. Quality Appraisal

The studies included in this review referred to various measures related to disaster preparedness and the response of home care providers. All studies had a descriptive, exploratory character, and most (i.e., five studies) used qualitative methods. Moreover, the only quantitative study [37], as well as the studies with a quantitative research component (mixed methods), remained on a descriptive level. Table 6 presents the methods and samples used in each study, along with the quality appraisal.

According to the MMAT, four studies fulfilled three or more of the five criteria. The data presented from these studies is easily comprehensible, has internal validity, and allows the research questions of the appraised study to be answered. Nonetheless, the external validity of the data is limited due to convenience sampling and existing selection bias. Four studies were rated, with fewer than three fulfilled MMAT criteria. The sampling and study designs used by these studies appeared to be appropriate to answer the research questions; however, the methodological procedures—such as the measurements and statistical analyses—were not easily comprehensible due to a lack of transparency, and the internal and external validity were thus difficult to assess. Furthermore, in studies with a mixed-method design, any inconsistencies or correlations between the methodological approaches that may have occurred were hardly mentioned. In one study, quality could not be assessed. Even though the three articles appeared to have scientific merit and may have pointed to guiding recommendations for home care providers to promote their disaster preparedness, there was a lack of information about the methodological approaches.

None of the examined studies investigated the effectiveness of the described disaster preparedness activities. Three studies evaluated tools (e.g., a disaster preparedness toolkit, a patient assessment tool, and a patient classification tool), albeit in terms of utility rather than effectiveness [37,38,39]. Almost all studies considered here took a primarily explorative and descriptive approach, which has inherently methodological limitations when it comes to measuring effectiveness. The included studies displayed strong heterogeneity with respect to different characteristics (which affects external validity) and indicated multiple initiatives on multiple levels. For example, some studies focused on the overall disaster preparedness of home care agencies (e.g., Ref. [33]), while other studies concentrated on a specific topic, such as patient classification (e.g., Ref. [39]). In addition to the fact that all of the studies included here were conducted in the USA, with a few exceptions, their data collection was regionally limited [33,37].

Individuals who reported information about the measures differed across studies in terms of their positions and affiliations with the surveyed home care providers (e.g., administrative staff, programme managers, practitioners, clinical staff, state-level preparedness experts). Similarly, differences existed between the health care settings and the types of providers included in the studies. For instance, while the respondents in some studies worked in home health and hospice agencies (e.g., Refs. [33,39], in other studies they worked in home-based primary care programmes [37,38]. Moreover, in some cases, personal care and home health care agencies were distinguished [34,35]. In addition to the lack of effectiveness evaluations, this high degree of heterogeneity impeded a comparison of measures across studies. 

## 4. Discussion

The aim of this integrative literature review was to obtain information on the implementation and effectiveness of organisational precautions that home care providers already take to prepare for disasters. 

Despite the rather limited literature on the topic, it became clear that there are a wide range of organisational precautions that home care providers have already taken or can take to strengthen their resilience to disasters. These precautions range from internal operational arrangements (e.g., the preparation of disaster plans) as well as internal and external information and networking activities (e.g., with patients, staff, or other health care providers in the community) to measures for safeguarding patient care (e.g., triage systems, evacuation precautions). Some of these precautions are innovative and inspiring and may have the potential to ensure decentralised care in times of disaster for as long as possible, thereby providing safety both for those in need of home care and for those who work in the field [13,15,16,17,18,19]. However, the literature does not provide any reliable information on whether some or all of these disaster preparedness arrangements are implemented systematically or consistently. Therefore, it remains unclear whether–and to what extent–people in need of home care or those responsible for the health and wellbeing of the population in a country can rely on these arrangements.

One of the main challenges of the present analysis stems from the fact that the included studies pointed to numerous initiatives at different levels—and as reported by various people—and that they used different survey methods. Due to their strong heterogeneity, the measures mentioned in the individual studies were hardly comparable either within or between studies. The predominantly regionally limited data collections as well as the methodological limitations of the primarily exploratory studies did not allow conclusions to be drawn about the systematic implementation of preparedness measures by home care providers. This finding was also shown in the literature review conducted by Wyte-Lake et al. [19] in 2012 and was confirmed ten years later in the present study. Consequently, no conclusions could be drawn about the actual dissemination or standardised use of measures beyond what the surveyed home care providers reported in each study, even for the USA, where these studies were undertaken. Even less can be said about the disaster preparedness arrangements of home care providers in other countries due to the lack of relevant research on the topic.

However, it should be considered whether the heterogeneity of the different measures described above is a characteristic of disaster preparedness among home care providers. In this case, they could be seen as an expression of the individual adaptation of the disaster preparedness measures to the very specific conditions on the site or in the respective service. This aspect should be adequately taken into account when comparing disaster planning measures and evaluating their effectiveness.

Although some of the included studies were methodologically well conducted, most were exploratory, descriptive, or comparative. There were hardly any evaluation or impact studies; thus, no statements can be made about which of the measures are effective and evidence-based. A few studies mentioned that home care providers reviewed their measures at regular intervals in the form of internal evaluation processes; however, scientifically evaluated data were not available. Thus, there appears to be a dearth of research on disaster preparedness measures for home care providers. This is perhaps not surprising given the paucity of robust empirical research in the field of home care overall, particularly when compared with inpatient and acute care [40,41]. This fact can be attributed to numerous problems, such as a lack of research expertise, capacities, and research funding in the field. Another reason for the lack of research that assesses the effectiveness of disaster preparedness measures in home care, in particular, might be that previous experience with disasters is highly important when it comes to disaster response [42]. The strength of qualitative research should be emphasised both because it can be used to answer questions about the experiences that home care providers have with disasters and because most of the included studies did so. In some studies, new measures were even implemented, or existing measures were adapted after lessons learned from disasters (e.g., [35]). However, the extent to which these experience-based measures result in better preparing home care providers for the next disaster remains unclear. This finding can pose significant challenges to policymakers, agencies, and providers alike when it comes to enacting or implementing effective disaster risk reduction measures. If the resilience of home care services as an infrastructure is to be strengthened, evidence-based planning and implementation are needed.

## 5. Limitations

The present review aimed to provide evidence of the effectiveness of primary studies that examined measures at the organisational level. Of the nine studies included in the review, only three received a 100% rating using the MMAT. Nevertheless, even if the desired findings on the evidence basis and the effectiveness of disaster preparedness measures were largely not achieved, the results of the present review provide a comprehensive overview of existing research activities on home care provider interventions and demonstrate the need for further evaluation studies. Such research is important in order to be able to target scarce resources in the future and direct these resources towards maintaining home care provision both during and after disasters.

Furthermore, we excluded articles that dealt only with educational aspects of disaster preparedness at the individual level. In this study, internal disaster training was considered as an outcome at the organisational level. In future research, external disaster preparedness teaching and training programs for health and care workers could also be included in the consideration.

Although no criterion was defined regarding the country in which the measures had to be implemented, all identified studies were from the USA, possibly because disasters occur more frequently in the USA than in other geographical regions (e.g., Europe). This fact may have painted a biased, less international picture of home care agencies´ disaster preparedness measures. Studies were also found from the Asia-Pacific region and Australia, both of which are particularly frequently affected by natural disasters [43,44,45,46], but these studies did not meet the inclusion criteria and were excluded due to their lack of focus on home care agencies or to their lack of a description of explicit measures at the organisational level. Additionally, only English-language articles were included, which may have excluded possible studies, for example, from the Asia-Pacific region. Furthermore, home care structures may exist in other (world) regions that make it difficult to implement concrete disaster-risk reduction measures where concrete measures for disaster preparedness are not explicitly located or have not yet been researched.

## 6. Conclusions

The purpose of the present integrative review was to present current evidence of disaster planning among home care providers. Although the research base on the topic raised here was highly unsatisfactory, the surveyed providers in the included studies appeared to be taking steps towards disaster preparedness. Such steps may serve as inspiration for countries in which this issue has played an even smaller role in home care than it has in the USA (e.g., in Germany or other European countries). However, there is a fundamental lack of evidence on how home care providers can design organisational disaster planning and, thus, how home care can be effectively maintained for vulnerable populations in the event of a disaster. Overall, home care should be moved more into the focus of health services research. Further research on disaster preparedness measures for home care providers is urgently needed, preferably with a systematic approach based on evaluation studies. Based on studies that provide valid data on the effectiveness of different disaster planning measures, systematic or even meta-analytic approaches could be used to identify best practises for home care providers with regard to disaster planning.

## Figures and Tables

**Figure 1 ijerph-20-05658-f001:**
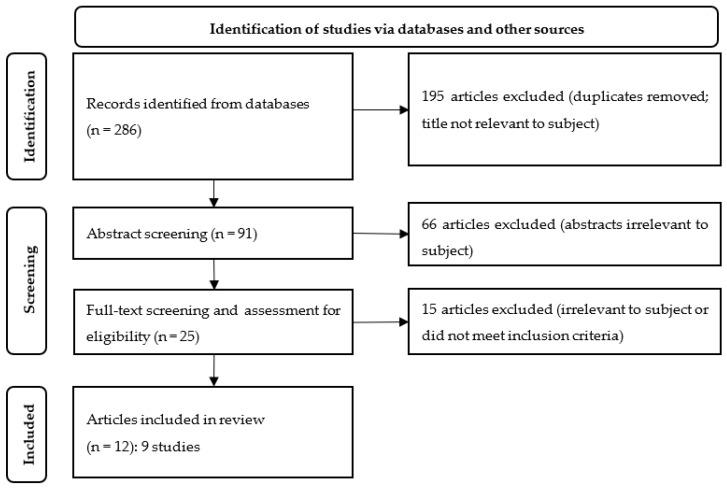
Flow chart with an overview of hits in specialised databases and other sources based on the 2020 PRISMA statement [27].

**Table 1 ijerph-20-05658-t001:** Search components and terms used in PubMed.

Search Components	Search Terms
Setting: Home care	home care services [MeSH Terms]; agency, home care [MeSH Terms]; home care [TIAB]; home nursing [TIAB]; home- and community-based service * [TIAB]; home-based primary care program * [TIAB]; HBPC [TIAB]; home health [TIAB]; domestic health care [TIAB]; domiciliary care [TIAB]
Intervention and Evaluation: Measures designed to help home care organisations improve disaster preparedness, and the outcome/evaluation of these measures	disaster planning [MeSH Terms]; emergency shelter [MeSH Terms]; disaster medicine [MeSH Terms]; emergency medicine [MeSH Terms]; civil defense [MeSH Terms]; disaster management [TIAB]; emergency management [TIAB]; disaster nursing [TIAB]; disaster role [TIAB]; disaster training [TIAB]; relief planning [TIAB]; evacuation [TIAB]; disaster evaluation [TIAB]; disaster mitigation [TIAB]; disaster preparedness [TIAB]; disaster response [TIAB]; disaster recovery [TIAB]; emergency preparedness [TIAB]; emergency response [TIAB]; disaster relief [TIAB]; disaster risk reduction [TIAB]

* stands for truncation.

**Table 2 ijerph-20-05658-t002:** Overview of the included articles.

Authors (Year); Origin	Title	Aims
ASPR TRACIE (2019) [33]; USA	Medical Surge and the Role of Home Health and Hospice Agencies (HHA)	To learn more about the implementation of the emergency management activities of Medicare-certified home health care and hospice agencies following the 2016 Final Rule of Centers for Medicare & Medicaid Services; to address these agencies’ (technical) assistance needs.
Balinsky 2003 [28]; Balinsky & Sturman 2006 [29]; Balinsky & Sturman 2006 [30]; USA	The Home Care Emergency Response to the September 11 Tragedy (1); Emergency Preparedness in the Home Care Environment–A Follow Up Study -Part I. Since 9/11, where has the urgency gone? (2); Emergency Preparedness in the Home Care Environment–A Follow Up Study -Part II. Since 9/11, where has the urgency gone? Are we really prepared? (3)	To examine the experiences of home care agencies in New York City and their losses after the 11 September attacks in order to develop recommendations (1); to examine the progress made in implementing the recommendations (2, 3).
Daugherty et al. (2012) [34]; USA	Disaster Preparedness in Home Health and Personal-Care Agencies: Are they ready?	To explore disaster preparedness policies and agency practices; to identify opportunities for coordination with disaster preparedness officials.
Kirkpatrick & Bryan (2007) [35]; USA	Hurricane Emergency Planning by Home Health Providers. Serving the Poor.	To gather critical information in order to help improve the response to community disasters that affect indigent populations, with a focus on emergency planning among home health care providers.
Laditka et al. (2008) [36]; USA	Disaster Preparedness for Vulnerable Persons Receiving In-Home, Long-Term Care in South Carolina.	To examine how agencies in South Carolina that provide in-home health care and personal care services help vulnerable clients prepare for disasters as well as how these agencies could improve their clients’ preparedness.
Wyte-Lake et al. (2017) [37]; USA	Developing a Home-based Primary Care Disaster Preparedness Toolkit	To examine the utility of an evidence-based disaster preparedness toolkit in the Veterans Health Administration’s (VHA) Home-Based Primary Care (HBPC) programme; to evaluate the toolkit.
Wyte-Lake et al. (2019) [38]; USA	Development of a Home Health Patient Assessment Tool for Disaster Planning	To develop a checklist-style tool that can be used to guide practitioners of the Veterans Health Administration’s (VHA) Home-Based Primary Care (HBPC) programme in assessing their patients’ disaster preparedness; to explore the utility of the tool’s implementation.
Wyte-Lake et al. (2020) [31]; Wyte-Lake et al. (2020) [32]; USA	Hurricanes Harvey, Irma and Maria: Exploring the Role of Home-based Care Programs (1) Preparedness and response activities of the US Department of Veterans Affairs (VA) home-based primary care (HBPC) program around the fall 2017 hurricane season (2)	To examine the activities of the nine VA HBPC programmes that were impacted during the 2017 Fall Hurricane Season; to develop a multi-layered understanding of what support the programmes need in the aftermath of hurricanes in order to better prepare their patients and staff for future disasters.
Zane & Biddinger (2011) [39]; USA	Home health Assessment Tools: Preparing for Emergency Triage	To develop a patient risk assessment tool that enables home care services, hospitals, and emergency planners to anticipate the care needs of all home care patients in a community in the event of a disaster.

**Table 3 ijerph-20-05658-t003:** Internal operational arrangements.

Study	Organisational Disaster Plans	Staff Disaster Arrangements	In-house Disaster Training
ASPR TRACIE (2019) [33]	✓		✓
Balinsky (2003) [28]; Balinsky & Sturman (2006) [29]; Balinsky & Sturman (2006) [30]	✓	✓	✓
Daugherty et al. (2012) [34]	✓		✓
Kirkpatrick & Bryan (2007) [35]	✓		✓
Laditka et al. (2008) [36]	✓	✓	
Wyte-Lake et al. (2017) [37]	✓		✓
Wyte-Lake et al. (2019) [38]			
Wyte-Lake et al. (2020) [31]; Wyte-Lake et al. (2020) [32]	✓		
Zane & Biddinger (2011) [39]			

**Table 4 ijerph-20-05658-t004:** Information and networking activities.

Study	Information Procedures			Networking
	Internal	For Home Care Users	External	
ASPR TRACIE (2019) [33]	✓	✓	✓	✓
Balinsky (2003) [28]; Balinsky & Sturman (2006) [29]; Balinsky & Sturman (2006) [30]	✓	✓	✓	✓
Daugherty et al. (2012) [34]		✓		✓
Kirkpatrick & Bryan (2007) [35]	✓	✓	✓	✓
Laditka et al. (2008) [36]	✓	✓		✓
Wyte-Lake et al. (2017) [37]				✓
Wyte-Lake et al. (2019) [38]		✓	✓	
Wyte-Lake et al. (2020) [31]; Wyte-Lake et al. (2020) [32]	✓	✓		✓
Zane & Biddinger (2011) [39]		✓	✓	

**Table 5 ijerph-20-05658-t005:** Measures for safeguarding patient care.

Study	Personal Patient Disaster Plan	Disaster Education	Triage/Patient Classification	Evacuation
ASPR TRACIE (2019) [33]	✓	✓	✓	✓
Balinsky (2003) [28]; Balinsky & Sturman (2006) [29]; Balinsky & Sturman (2006) [30]			✓	
Daugherty et al. (2012) [34]		✓	✓	✓
Kirkpatrick & Bryan (2007) [35]		✓		✓
Laditka et al. (2008) [36]	✓	✓	✓	✓
Wyte-Lake et al. (2017) [37]				
Wyte-Lake et al. (2019) [38]	✓	✓	✓	✓
Wyte-Lake et al. (2020) [31]; Wyte-Lake et al. (2020) [32]	✓	✓	✓	✓
Zane & Biddinger (2011) [39]			✓	✓

**Table 6 ijerph-20-05658-t006:** Overview of the methods, samples, and quality appraisal of the included articles.

Authors (Year); Origin	Methods/Sample	Quality Appraisal (MMAT)
ASPR TRACIE (2019) [33]; USA	Mixed-methods design; quantitative: online survey of leaders of Medicare-certified home health and hospice agencies from 43 states (n = 245); qualitative: semi-structured in-depth telephone interviews (n = 25)	5/5 criteria met; self-selection bias
Balinsky (2003) [28]; Balinsky & Sturman (2006) [29]; Balinsky & Sturman (2006) [30]; USA	Follow-up survey with qualitative interviews; roundtable discussion; home care agencies (n = 8) in Lower Manhattan that were impacted by the 11 September attacks (follow-up: n = 6); roundtable discussion: the 8 home care agencies as well as representatives from government, academia, and other home care agencies (n = unknown)	0/5 criteria met; insufficient indication of data sampling, data collection detail, data analysis, and data consistence
Daugherty et al. (2012) [34]; USA	Qualitative; semi-structured interviews; administrators (n = 21) of home health and personal care agencies in Georgia and Southern California	5/5 criteria met; self-selection bias
Kirkpatrick & Bryan (2007) [35]; USA	Qualitative; case study approach; retrospective; in-depth interviews; the two top administrative staff of 5 selected home health agency facilities that operate in Orleans Parish (n = 10?)	2/5 criteria met; insufficient indication of study sampling, data collection detail, and analysis
Laditka et al. (2008) [36]; USA	Qualitative; semi-structured interviews; state-level preparedness experts (n = 9), administrators of home health (n = 5), and in-home personal care agencies (n = 16) in South Carolina	2/5 criteria met; insufficient indication of study sampling, data collection detail, and analysis
Wyte-Lake et al. (2017) [37]; USA	Quantitative online survey; programme managers of Veterans Health Administration Home-Based Primary Care (VHA HBPC) programmes across the country (n = 77)	4/5 criteria met; insufficient indication of non-response analysis
Wyte-Lake et al. (2019) [38]; USA	Mixed-methods design; quantitative: online survey of practitioners (n = 64) at 10 sites of (VHA HBPC) programmes in 8 states as a natural cohort from an existing study that used the tool and completed patient questionnaires (n = 754); follow-up survey with practitioners (n = 33) of 2 sites; qualitative: follow-up feedback interviews with programme manager (n = unknown)	5/5 criteria met; self-selection bias
Wyte-Lake et al. (2020) [31]; (Wyte-Lake et al. (2020) [32]; USA	Qualitative (1); mixed-methods design (2); qualitative: semi-structured interviews; clinical staff (n = 34) with key functions of the 9 Veterinary Affairs (VA) HBPC from Texas, Florida, and Puerto Rico (1,2); quantitative: secondary data analysis; timeline of activities of the same 9 sites that use the VA’s Corporate Data Warehouse (CDW) (2)	2/5 criteria met; insufficient indication of data collection detail, data analysis, and data consistency
Zane & Biddinger (2011) [39]; USA	Qualitative; unstructured interviews; home health and hospice agencies (n = 21) and other community service providers (n = 4) from 6 states	2/5 criteria met; insufficient indication of data collection detail and data analysis

## Data Availability

No new data were created or analyzed in this study. Data sharing is not applicable to this article.

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
