# Peer review of "Evidence of Disaster Planning by Home Care Providers: An Integrative Literature Review"

_ijerph, 2023, doi:10.3390/ijerph20095658_

Round 1

Reviewer 1 Report

The article provides a general overview of the studies reviewed, but it could benefit from providing more specific details on the research methods and results of each study. This would help readers understand the strengths and limitations of each study and provide more insight into the overall evidence base. i.e Table as a appendix

You concluded that evidence on how to make organizational disaster planning effective and sustainable remains lacking. However, it could benefit from a more detailed discussion of the implications of this finding, such as the potential impact on policy and practice, and suggestions for future research.

In discussion 1st paragraph should be moved to the end of the introduction

As noted in the article, none of the studies investigated the effectiveness of disaster planning by home care providers. The authors could consider conducting a systematic review or meta-analysis to investigate the effectiveness of different disaster planning strategies and identify best practices for home care providers.

Reviewer 2 Report

Thanks to the editors for the opportunity to review this review article.

I think that this subject is very important, especially defining what preventive measures are of the most iportance for decentralised services in a home care settings.

The article is written sistematically and it is concise. Figure and tables are underastandable and informational.

It is not obligatory reccomendation, but I would just suggest to authors to include information about how this topic (disaster planning and preventive measures by home care providers) is frequent in a curriculums at medical and nursing schools. Maybe, if it is not present enough in the curriculums in schools, the authors could do something in the sense of recommending it in a means of importance of this subject and learning about it in medical and nursing schools. This is important because, the nurses and doctors should be prepared for this kind of challenge when they graduate.

I enjoyed reading this review and I hope the authors will continue to do the research in this topic, because it is not present enough in literature.

Reviewer 3 Report

This manuscript examines existing literature on disaster planning by home care providers. Using an integrative literature review, the authors identified nine different original research studies that were analyzed in an in-depth manner. The manuscript focuses on an important topic – planning among home care providers, which is an underexamined area. It is well-written, and the methods section is very detailed. However, some sub-sections of the results section are brief and need some elaboration.  Currently, the authors only summarize the specific contents within the selected research studies. The manuscript could be improved by using examples from the articles to support the findings. For instance, under 3.3.1 -Organizational disaster plan, use examples from some of the selected articles to elaborate on different types of plans. The same can be done for each subsection to make the results section stronger.

Below are some other suggestions to improve the manuscript:

·         Under the Introduction section, also discuss the home-bound population or special needs population as a vulnerable population and why this topic is important.

·         In lines 191,199 and 207, the sub-headings  “Spreading information…” can sound better if revised as “Disseminating information…”

·         Under the Discussion section, provide the study implications for theory and practice.

·         Under the Conclusion section, provide some more future research recommendations. These could include research recommendations based on the study's limitations.

Reviewer 4 Report

Dear authors,

Thank you for your interesting contribution. Here are some of my suggestions.

Line 149

3.3. Thematic Analysis

Regarding the Whittemore & Knafl review method, I admit that I am new to this method, but since this is a literature review, I recommend inserting the citation number even in the document describing the subcategories. That is, the data were not obtained through interviews, so please identify the supporting literature, even for subcategories.

Lines 318331

In this paragraph, you are referring to the practical dissemination or standardized use of measures, considering the weather conditions, geographical features, disaster characteristics, etc. of each country, it is rather natural that standardization is difficult. Also, the definition of home care providers varies from country to country. Rather, it can be said that the lack of standardization is a characteristic of disaster countermeasures by home care providers.

I recommend that you reconsider this paragraph.

Line 351

5. Strengths and Limitations

Please write only limitations. If you want to emphasize Strengths, I recommend writing it in the discussion.
